# Quantifying Effects of Earth Orbital Parameters and Greenhouse Gases on Mid-Holocene Climate

Yibo Kang and Haijun Yang*

Department of Atmospheric and Oceanic Sciences and Institute of Atmospheric Science and CMA-FDU Joint Laboratory of Marine Meteorology, Fudan University, Shanghai, 200438, China.

Shanghai Scientific Frontier Base for Ocean-Atmosphere Interaction Studies, Fudan University, Shanghai 200438, China.

*Correspondence to*: Haijun Yang (yanghj@fudan.edu.cn)

**Abstract.** The mid-Holocene (MH) is the most recent typical climate period, and a subject of great interest in global paleocultural research. Following the latest Paleoclimate Modelling Intercomparison Project phase 4 (PMIP4) protocol and using a fully coupled climate model, we simulated the climate during both the MH and the pre-industrial (PI) periods, and quantified the effects of Earth orbital parameters (ORB) and greenhouse gases (GHG) on climate differences, focusing on the simulated differences in the Atlantic meridional overturning circulation (AMOC) between these two periods. Compared to the PI simulation, the ORB effect in the MH simulation led to seasonal enhancement of temperature, consistent with previous findings. In the MH simulation, the ORB effect led to a markedly warmer climate in the mid-to-high latitudes and increased precipitation in the Northern Hemisphere, which were partially offset by the cooling effect of the lower GHG. The AMOC in the MH simulation was about 4% stronger than that in the PI simulation. The ORB effect led to 6% enhancement of the AMOC in the MH simulation, which was, however, partly neutralized by the GHG effect. Transient simulation from the MH to the PI further demonstrated opposite effects of ORB and GHG on the evolution of the AMOC during the past 6000 years. The simulated stronger AMOC in the MH was mainly due to the thinner sea ice in the polar oceans caused by the ORB effect, which reduced the freshwater flux export to the subpolar Atlantic and resulted in a more saline North Atlantic. This study may help us quantitatively understand the roles of different external forcing factors in Earth's climate evolution since the MH.

**Keywords**: Mid-Holocene, Earth orbital parameters, Greenhouse gases, Atlantic Meridional Overturing Circulation

**1. Introduction**

The mid-Holocene (MH; 6000 years before the present) is a period of profound cultural transition worldwide, particularly in the arid-semi-arid belt of ~30°N (Sandweiss et al., 1999; Moss et al., 2007; Roberts et al., 2011; Warden et al., 2017). The MH climate, which belongs to the Holocene climatic optimum (Rossignol-Strick, 1999; Chen et al., 2003; Zhang et al., 2020),  differs notably from that of the subsequent period. Many studies have shown that the development of human civilization during this period was influenced by the climate, which was closely related to external factors such as the Earth's orbital parameters (ORB), greenhouse gases (GHG), and solar constants (Jin, 2002; Wanner et al., 2008; Warden et al., 2017). Therefore, it is of great interest to study the MH climate, for a better understanding of the influence of external forcing factors on human civilization.

As the key benchmark period of the Paleoclimate Modelling Intercomparison Project (PMIP) program (Joussaume and Taylor, 1995; Kageyama et al., 2018), the MH experiment was designed to examine climate response to a change in the seasonal and latitudinal distribution of incoming solar radiation caused by known changes in Earth orbital forcing. As the program evolved, the GHG concentrations used in the MH experiments are closer to the true values (Monnin et al., 2001, 2004). However, most studies focused on the general climate differences between the MH and pre-industrial (PI) periods; the individual effects of the ORB and GHG on the climate itself are not isolated. Some studies examined the role of GHG by comparing different PMIP programs. Otto-Bliesner et al. (2017) found that the change in the experimental protocol between PMIP phase 4 and PMIP phase 3 (PMIP4 and PMIP3 hereafter, respectively), with a reduction in $CO_2$ concentration from 280 to 264.4 ppm, would reduce GHG forcing by about 0.3 W/m$^{-2}$. This change can produce an estimated global mean cooling in surface air temperature (SAT) of about 0.28℃ based on the climate sensitivity of difference models in PMIP4 (Brierley et al., 2020). The GHG contribution to temperature change is small, but not negligible. Quantifying the effects of ORB and GHG on the difference between the MH and PI has important implications for a deeper understanding of the roles played by external forcing factors in the past climate.

The Atlantic meridional overturning circulation (AMOC) is considered an important heat transmitter of the Earth's climate system, which affects global climate on various timescales (Rahmstorf, 2006). Paleoclimate studies showed that the weakening or stopping of the AMOC can result in substantial cooling across the Northern Hemisphere (NH) (Brown and Galbraith, 2016; Yan and Liu, 2019). In recent years, predictions concerning future behavior of the AMOC by the Intergovernmental Panel on Climate Change (IPCC) are accompanied by notable uncertainties, particularly due to the substantial variability in anticipated AMOC changes under different emission scenarios (Fox-Kemper et al., 2021). Therefore, simulating past AMOC changes and exploring the effects of different forcing factors

on its behavior will help us understand the nature of abrupt climate change in the past and mitigate uncertainties in
future climate projections. In the previous MH simulations of the PMIP, the AMOC was generally stronger than that
of the PI (Găinuşă-Bogdan et al., 2020); this change in the AMOC is related to sea ice feedback, and the simulation
results may be slightly different due to model or resolution differences (Shi and Lohmann, 2016; Shi et al., 2022).
Recent studied suggested that the difference of the AMOC between the MH and PI periods in PMIP4 ensemble
simulation is not significant (Brierley et al., 2020). By comparing the strength of the AMOC during the interglacial
period, it was found that the variation range of the AMOC in the MH is within the internal variability range of all
models; and the ORB does not seem to have played a role (Jiang et al., 2023a). By examining multi-model transient
simulations that all include two or more external forcing factors, Jiang et al. (2023b) reported that the AMOC did not
change much from the MH to the PI, which is consistent with some proxy reconstructions.
In this paper, we further study the mechanism of weak difference of the AMOC between the MH and PI periods.
The effects of different external forcings on the AMOC are quantified through several sensitivity experiments.
Multiple transient experiments are also performed to verify the roles of different forcing factors in long-term climate
evolution. This paper is structured as follows. An introduction to the fully coupled climate model is given in section 2,
along with experimental design. In section 3, we present the effects of ORB and GHG on the MH climate, and their
effects on the Hadley cell and AMOC. The changes of North Atlantic Ocean buoyancy between the MH and PI
periods in both equilibrium and transient experiments are described in section 4. Summary and discussion are given in
section 5.

**2. Model and experiments**
The coupled model used in this study is the National Centre for Atmospheric Research's Community Earth
System Model version 1.0 (CESM1.0). It includes atmospheric, oceanic, sea-ice, and land model components. The
atmospheric model consists of 26 vertical levels and T31 horizontal resolution (roughly 3.75°×3.75°). The land model
shares the same horizontal resolution as the atmospheric model. The ocean model has 60 vertical levels, and employs
gx3v7 horizontal resolution. In the zonal direction, the grid has a uniform 3.6° spacing. In the meridional direction, the
grid is nonuniformly spaced: it is 0.6° near the equator, gradually increases to the maximum 3.4° at 35°N/°S, and then
decreases poleward. The sea-ice model has the same horizontal resolution as the ocean model. More details on these
model components can be found in a number of studies (Smith and Gent, 2010; Hunke and Lipscomb, 2010; Lawrence
et al., 2012; Park et al., 2014).

86       To quantify the effects of ORB and GHG on climate differences between the MH and PI periods, we designed

three sensitivity experiments following the PMIP4 protocol (Table 1). Exp MH uses the ORB and GHG in the MH
period. Exp MH_ORB uses the ORB in the MH period and the GHG in the PI period. Exp PI uses the ORB and GHG
in the PI period. Note that our simulations do not intend to compare climate states between PMIP3 and PMIP4; we
want to isolate the individual effects of ORB and GHG within the framework of the PMIP4. There are differences
between PMIP3 and PMIP4 in solar constant and GHG concentration. By tightly controlling the external forcings in
the different experiments, our simulations effectively isolate the external forcing component compared to PMIP3, not
just the ORB. The solar constant in the three experiments is set to 1360.75 $W/m^2$. The specific values of the ORB are
listed in Table 1 (Berger and Loutre, 1991); and the GHG data comes from the ice-core records of the Antarctica and
Greenland (Otto-Bliesner et al., 2017). The vernal equinox is set to noon on 21 March. Exps MH and MH_ORB start
from the PI condition, and each of the three experiments is integrated for 2000 years and reaches the equilibrium by
then (Fig. 5a). The effect of ORB is obtained by subtracting Exp PI from Exp MH_ORB, and the effect of GHG is
obtained by subtracting Exp MH_ORB from Exp MH. The combined effect of ORB and GHG is obtained by
subtracting Exp PI from Exp MH. In this paper, we use the monthly mean data of the last 500 years of each model
simulation for analysis (Fig. 5a).
To enhance the rigor of our study and confirm the effects of ORB and GHG on the climate evolution from the
MH to the PI, we conducted three additional transient experiments (Table 2). Each transient experiment starts at the
MH and concludes at the PI, spanning a total of 5900 model years. Exp ORB represents the transient experiment for
ORB; Exp GHG, the transient experiment for GHG; and Exp Full, the experiment where ORB, GHG, and total solar
irradiance are applied concurrently. The ORB data in the transient experiments is from Berger and Loutre (1991), the
GHG data is interpolated from GHG data reconstructed from Antarctic ice cores, and the total solar irradiance data is
from the PMIP4 SATIRE-M solar forcing data (Otto-Bliesner et al., 2017). We use model years 1–500 to represent the
MH climate (Stage 1) and model years 5401–5900 to represent the PI climate (Stage 2), and then compare the
difference between Stage 1 and Stage 2 to the results of the equilibrium experiments (Fig. 5b). The settings for forcing
information in the transient experiments are listed in Table 2.

**Table 1. Forcings and boundary conditions in equilibrium experiments. More details can be found in Otto-Bliesner et al.**
**(2017).**

| | Exp MH | Exp PI | Exp MH_ORB |
|---|---|---|---|
| Orbital parameters | | | Same as Exp MH |
| Eccentricity | 0.018682 | 0.016764 | 0.018682 |

| | | | |
|---|---|---|---|
| Obliquity (degrees) | 24.105 | 23.459 | 24.105 |
| Perihelion – 180 | 0.87 | 100.33 | 0.87 |
| Greenhouse gases | | | Same as Exp PI |
| $CO_2$ (ppm) | 264.4 | 284.3 | 284.3 |
| $CH_4$ (ppb) | 597 | 808.2 | 808.2 |
| $N_2O$ (ppb) | 262 | 273.0 | 273.0 |


**Table 2. Forcing and boundary conditions in transient experiments.**

| | Exp ORB | Exp GHG | Exp Full |
|---|---|---|---|
| Orbital parameters | Berger and Loutre (1991) | Same as Exp MH | Same as Exp ORB |
| Greenhouse gases | Same as Exp MH | Flückiger et al. (2002) Monnin et al. (2004) Spahni et al. (2005) | Same as Exp GHG |
| Total solar irradiance | Same as Exp MH | Same as Exp MH | Otto-Bliesner et al. (2017) |


Orbital parameters include eccentricity, precession, and obliquity. In the past six millennia, both eccentricity and
obliquity did not change much. The main change came from precession, which is influenced by eccentricity and the
longitude of perihelion. As a result, perihelion is close to the NH autumn equinox in the MH period and close to the
NH winter solstice in the PI period. Therefore, with respect to Exp PI, the solar energy received at the top of the
atmosphere (TOA) in Exp MH changed seasonally and latitudinally, as shown in Fig. 1a. Compared to Exp PI, Exp
MH had higher NH summer radiation and lower winter radiation, and the difference during June–August (JJA)
reached 30 W/m$^2$ in the high latitudes. Smaller precession led to more radiation received in the NH summer in the MH
period. Figure 1b shows the meridional variation of annual mean shortwave radiation at the TOA, which is greater
than 4 W/m$^2$ poleward of 45°N(S), but negative and smaller than 1 W/m$^2$ between 45°S and 45°N. This situation is
associated with the larger obliquity in the MH (Otto-Bliesner et al., 2006; Williams et al., 2020). In addition, the
difference of GHG between the MH and PI periods can lead to an effective radiative forcing of 0.3 W/m$^2$ (Otto-
Bliesner et al., 2017).

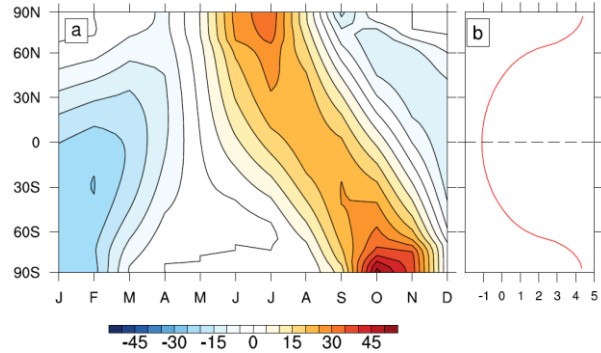


**Figure 1   (a) Latitude-month distribution of solar radiation change at the TOA in Exp MH, and (b) annual mean solar**
**radiation change, with respect to Exp PI. Units: W/m².**

## 3.  Results

### 3.1 Surface air temperature and precipitation

Compared to Exp PI, Exp MH has warmer annual mean temperatures in the NH high latitudes and cooler
temperatures in the rest of the globe (Fig. 2a), while Exp MH_ORB has a warmer surface at mid–high latitudes in both
the NH and SH, with a greater range and magnitude than Exp MH (Fig. 2b). Figure 2b shows direct response to the
meridional change of annual mean solar radiation. The lower GHG in the MH contributed to a lower global surface
temperature, which is clear in the mid–high latitudes (Fig. 2c). In the NH summer (June–August, or JJA), Exp MH
shows a general warming of more than 1℃ north of 30°N, which is more significant in Greenland and Euro-Asian
continent, and a cooling belt in northern India and central Africa (Fig. 2d), which is associated with increased rainfall
due to the enhanced monsoon (Fig. 2d). The magnitude and extent of warming due to the ORB effect are apparently
greater, with warming of up to 3℃ in central Asia (Fig. 2e). The GHG cooling is more pronounced over the Southern
Ocean (Fig. 2f). In the NH winter (December–February, or DJF), only the NH polar latitudes remain the warming.
There is strong cooling (up to 3℃) in the African and Euro-Asian continents (Fig. 2g). The patterns under the ORB
and GHG forcing are similar to their annual mean situations, except for the enhanced cooling in South Asia and
central Africa (Fig. 2h) and over the subpolar Atlantic (Fig. 2i). Most figures show polar amplification, which may be
related to the change of sea ice (Otto-Bliesner et al., 2017; Williams et al., 2020).

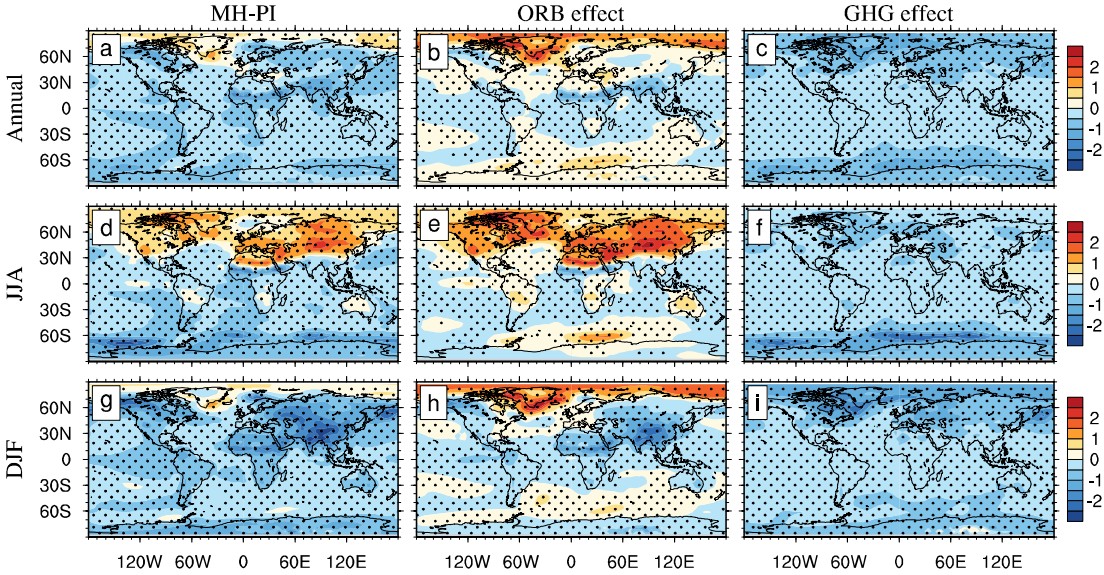


**Figure 2 (Left column) Changes in SAT in Exp MH, with respect to Exp PI, and the contributions from (central column)
the ORB effect and (right column) the GHG effect. (a)–(c) are for annual mean; (d)–(f), for NH JJA; and (g)–(i), for NH
DJF. Stippling shows significance over the 90% level calculated by Student *t*-test. Units: ℃.**


Differences in precipitation between the MH and PI simulations are shown in Fig. 3. Consistent with the

latitudinal and seasonal differences of insolation (Fig. 1), the largest difference in precipitation between the two

periods also occurs in the NH summer, with significantly more precipitation in northern India and equatorial African

monsoon region, and drier in the equatorial Atlantic and Pacific in Exp MH (Fig. 3d). The difference between Exps

MH and PI is mainly in the global tropics, and is contributed predominantly by the ORB effect (Figs. 3e, h), as the

GHG effect is very weak (Figs. 3f, i).

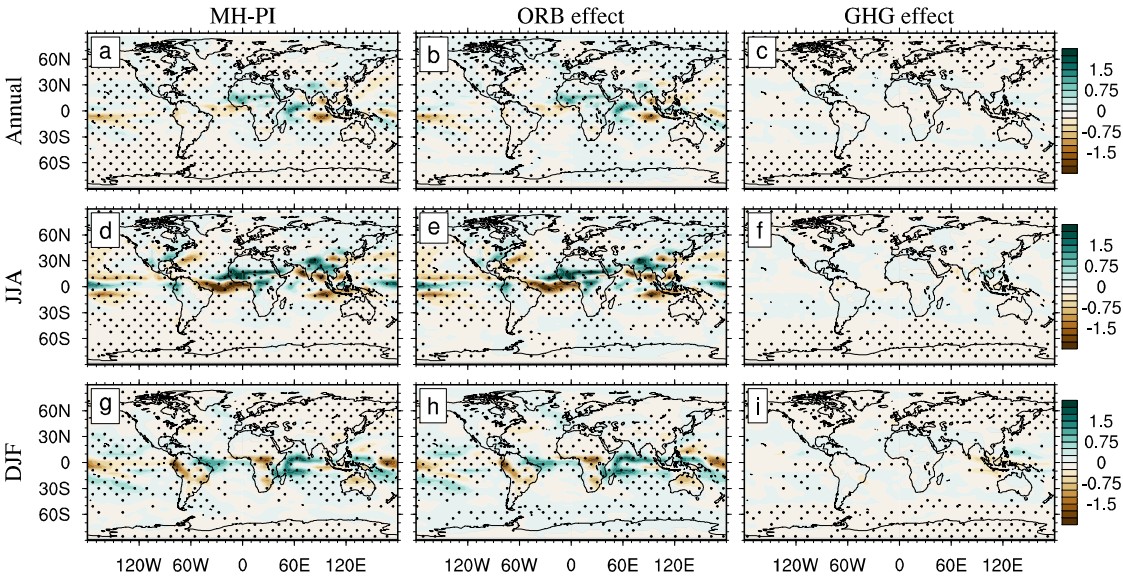

159

**Figure 3   Same as Fig. 2, but for precipitation change. Units: mm/day.**

Although the numerical values may be slightly different due to different models or resolutions, in general the annual and seasonal climatology differences of temperature and precipitation between Exps MH and PI are in good agreement according to recent studies (Williams et al., 2020; Zhang et al., 2021b). The ORB effect dominates the changes in global surface temperature and precipitation. Thus, Exp MH has a warmer climate than Exp PI, particularly in NH high latitudes.

## 3.2 Meridional atmospheric circulation

The meridional atmospheric circulation, namely, the Hadley cell, in Exp MH is about 10% weaker than that in Exp PI (Fig. 4a), consistent with the weaker meridional atmospheric temperature gradient in Exp MH than in Exp PI. The weaker Hadley cell in Exp MH is mainly due to the ORB effect (Figs. 4b, e, h). The GHG effect can be neglected (Figs. 4c, f, i). The Hadley cell is weaker due to the strong warming of the high-latitude temperatures in the NH summer (Fig. 4d). The strengthening of the Hadley cell in the NH winter (Fig. 4g) corresponds to an increasing temperature gradient between the tropics and mid latitudes (Fig. 2g). The weaker Hadley cell also leads to a weaker meridional atmospheric heat transport from low to high latitudes, which will be discussed in section 3.4.

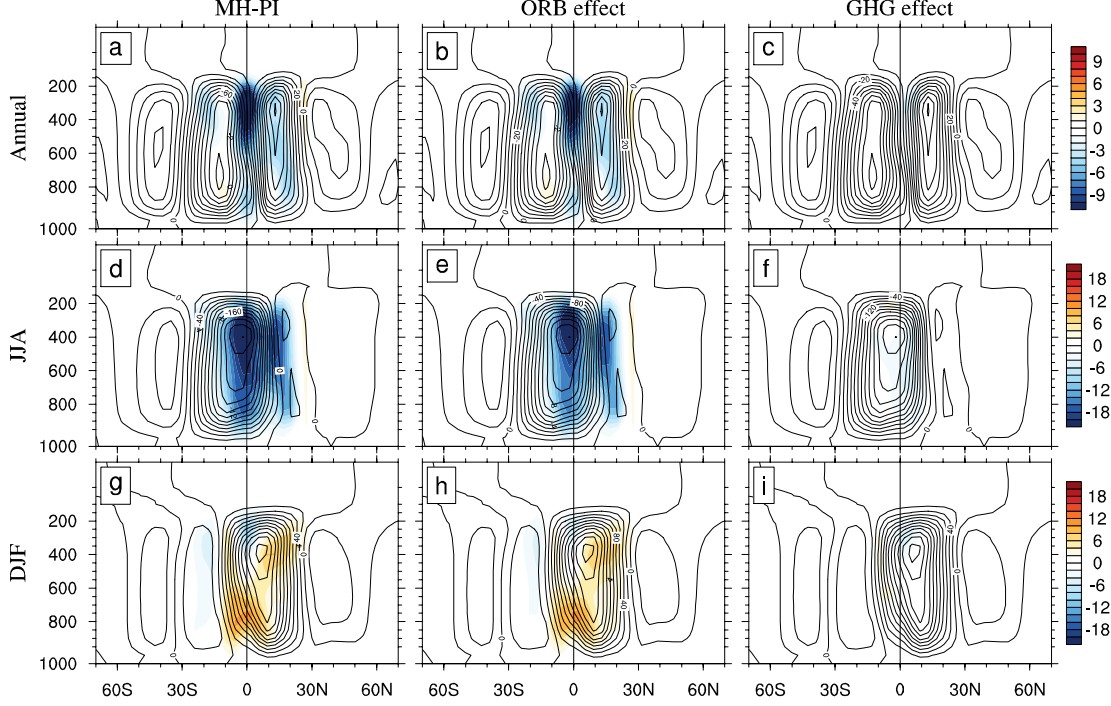

**Figure 4   Same as Fig. 2, but for the mean Hadley cell in Exp PI (contour) and its changes (shading) in Exp MH. Units: $10^9$ kg/s.**


## 3.3 Atlantic meridional overturning circulation


The AMOC strength, defined as the maximum streamfunction between 0 and 2000 m and between 20° and 70°N
in the North Atlantic, are 19.4 and 18.3 Sv in Exps MH and PI, respectively. Figure 5a shows the time series of the
AMOC of the three equilibrium experiments, all of which reached the equilibrium state. The AMOC in Exp MH_ORB
(dark blue line) is 1 Sv stronger than that in Exp PI (dark red line), while the AMOC in Exp MH (dark black line) is
roughly the same as that in Exp MH_ORB. Figure 5b shows the evolution of the AMOC in the three transient
experiments. In Exp ORB, the AMOC strength shows a downward trend (dark blue line). In Exp GHG, the AMOC
strength exhibits a slight increase with an indistinct trend (dark red line). In Exp Full, the trend of AMOC strength is
essentially between Exps ORB and GHG, indicating a combined effect of external forcing factors (dark black line).

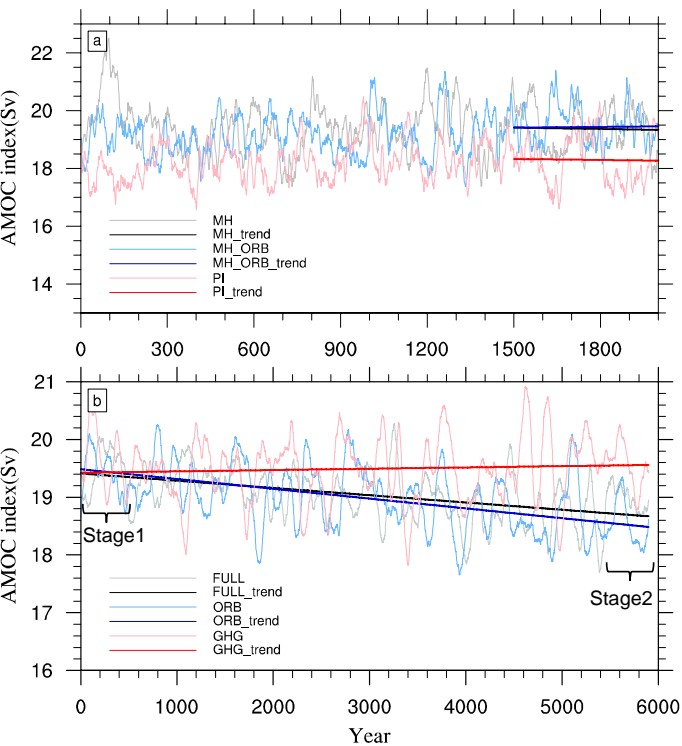


**Figure 5   (a) Evolutions of the AMOC in Exp MH (gray and black lines), Exp MH_ORB (blue lines), and Exp PI (red lines).**
**(b) Evolutions of the AMOC in Exp Full (gray and black lines), Exp ORB (blue lines), and Exp GHG (red lines). The thick**
**lines indicate the linear trends of the AMOC in different experiments. Units: Sv (1 Sv=$10^6$ m$^3$/s).**

The patterns of the AMOC are shown in Fig. 6; the depth of the maximum AMOC in all experiments occurs near
1000 m. The AMOC patterns in Exps MH and PI are similar (Figs. 6a, c), which suggests that the combined effect of
the ORB and GHG on the AMOC is small (Fig. 6d). This is similar to some recent studies, even though there are
slight differences north of 45°N (Brierley et al., 2020; Williams et al., 2020). Individual effects of the ORB and GHG
are not negligible (Figs. 6e, f). In fact, the ORB effect leads to 6% stronger AMOC in Exp MH than in Exp PI (Fig.
6e). The deep overturning is significantly enhanced south of 45°N, but slightly weakened north of 45°N. However, at
the same time the GHG effect leads to a slight decline in AMOC strength in Exp MH, especially above 1500 m south
of 45°N (Fig. 6f). The ORB and GHG have opposite effects on the AMOC, which make the AMOC in Exp MH
roughly the same as that in Exp PI. Figure 6g-i further shows the effects of different forcing factors on the AMOC
patterns in the transient experiments, which are similar to the changes in the equilibrium experiments (Figs. 6d–f),
although there are differences in intensity. The offset effect between ORB and GHG in the transient experiments is the
same as that in the equilibrium experiments. Some scholars have suggested that the change of AMOC in Exp MH may
come from internal variability (Williams et al., 2020), but it is clear from our simulations that changes in response to
external forcings are the main reason for the variations that occur in Exp MH.

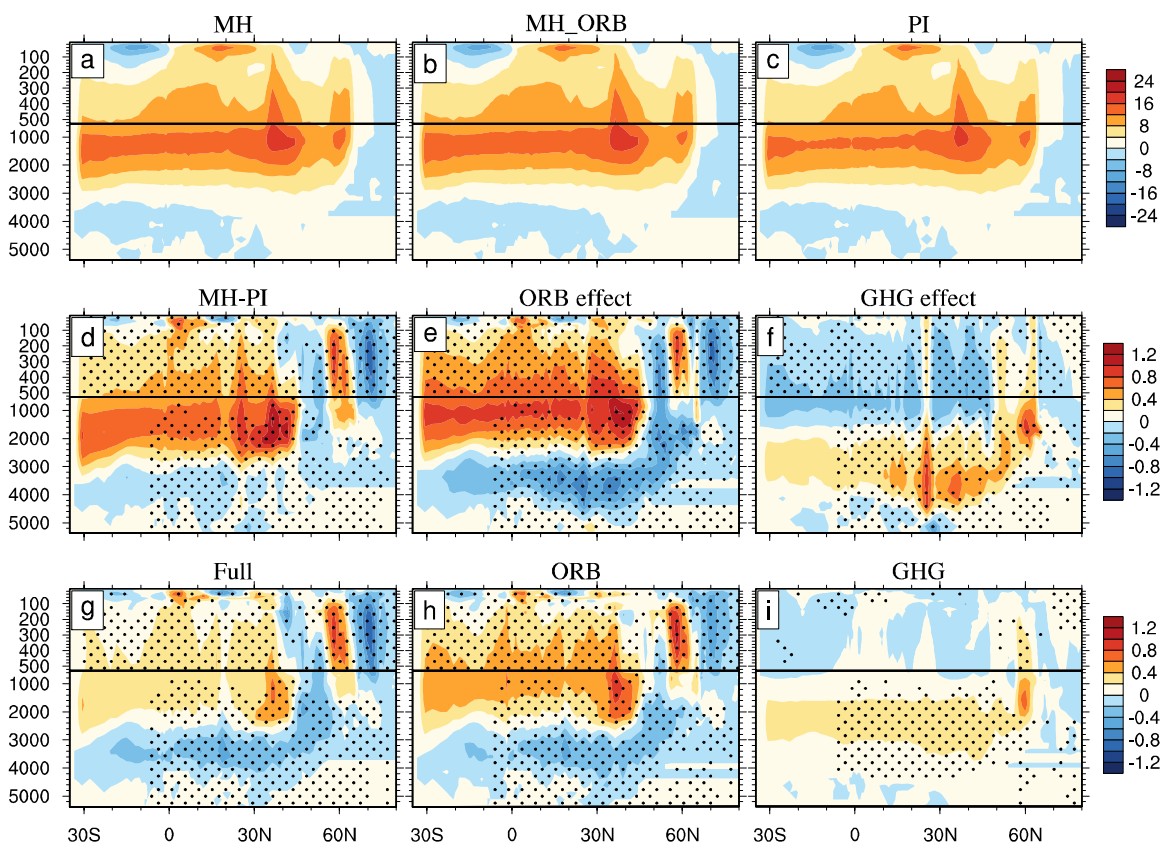

**Figure 6   Patterns of mean AMOC in (a) Exp MH, (b) Exp MH_ORB, and (c) Exp PI; and (d) AMOC change in Exp MH,**
**with respect to Exp PI. (e) and (f) show AMOC changes due to the ORB effect and GHG effect, respectively. g, h, i represent**
**the AMOC changes between the two stages (Stage1-Stage2) in Exps Full, ORB, and GHG, respectively. The AMOC index is**
**defined as the maximum streamfunction in the range of 0–2000 m of 20°–70°N in the North Atlantic. Stippling shows**
**significance over the 90% level calculated by Student t-test. Units: Sv.**

**3.4 Meridional heat transport**

Meridional heat transport (MHT) plays an important role in maintaining energy balance of the Earth climate system. Figure 7a shows the annual MHTs in different experiments, which are nearly identical. The climate differences between Exps MH and PI hardly change the integrated heat transport in both the atmosphere and ocean. Consistent with previous studies (Trenberth and Caron, 2001), the annual mean MHT shows an antisymmetric structure about the equator, with the peak value of about 5.5 PW (1 PW=$10^{15}$W) at 40°N/S. Compared with ocean heat transport (OHT), the atmosphere heat transport (AHT) dominates at most latitudes, which is also consistent with previous studies (Held, 2001; Wunsch, 2005; Czaja and Marshall, 2006).

However, the MHT changes caused by the ORB and GHG effects appear to be nonnegligible. The ORB causes an increase in OHT in the NH, with the maximum change of about 0.10 PW near the equator, roughly 10% of the mean OHT there. This is due to the enhanced AMOC, and is the main cause of temperature increase in the NH high latitudes (Fig. 2b). The northward AHT is reduced, with the maximum change of about 0.10 PW. This is due to the weakend Hadley cell. The AHT change compensates the OHT change very well in the deep tropics, while the former overcompensats the latter in the NH off-equatorial regions (Fig. 7b). The GHG effect on the MHT is very weak, with the maximum MHT change of no more than 0.04 PW near 5°N (Fig. 7c), which is just one third of the ORB-induced MHT change (Fig. 7b).

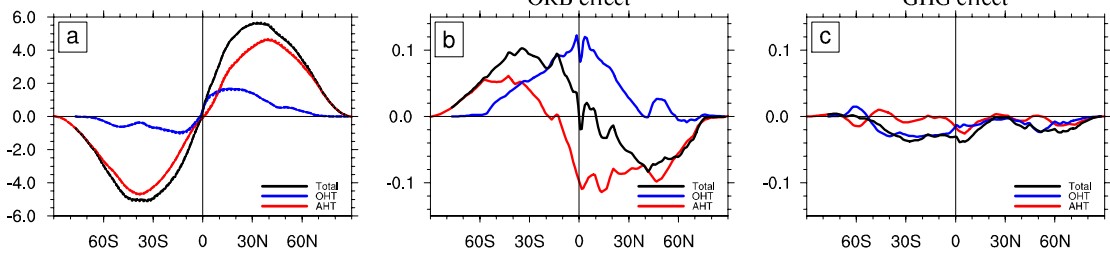

**Figure 7** **(a) Annual mean meridional heat transport (MHT). Black, red, and blue lines are for the total MHT, AHT, and OHT, respectively. Solid, dashed, and dotted lines are for Exps MH, MH_ORB, and PI, respectively. (b) and (c) show changes in the total MHT, AHT, and OHT due to ORB and GHG effects, respectively. Units: PW (1 PW = $10^{15}$ W).**

**4.  Changes in North Atlantic Ocean**

**4.1 Changes in sea-surface temperature, salinity, and density**

The strength of the AMOC is largely determined by the North Atlantic deep-water formation, which is in turn determined by upper-ocean density. Figures 8 and 9 show the differences of sea-surface temperature (SST), salinity

(SSS), and density (SSD) in the North Atlantic between Exps MH and PI, and the two stages in the transient
experiments, respectively. The SST difference is characterized by a warming up to 1.6℃ in the subpolar Atlantic and a
cooling of about 1℃ near the Nordic Seas and Gulf Stream extension region (Fig. 8a). The surface ocean warming in
the North Atlantic is due to the ORB effect (Fig. 8b), which causes a strong and extensive warming in the North
Atlantic, with the maximum warming in the subpolar Atlantic. This is in contrast to the warming hole shown by
observations, which are dominated by the cooling of the North Atlantic in the context of global warming. The GHG
effect causes a general cooling in the North Atlantic (Fig. 8c), offsetting partially the ORB-induced warming, leaving
a cooling in the Nordic Seas and Gulf Stream extension (Fig. 8a). The North Atlantic SST change in Exp Full (Fig. 9a)
is consistent with that of Exp MH (Fig. 8a), although the magnitude is slightly smaller. Exp ORB also exhibits
stronger warming than Exp Full (Fig. 9b), consistent with Fig. 8b. Exp GHG shows a slight cooling (Fig. 9c),
consistent with Fig. 8c. Overall, the SST change in the transient experiments is the same as that in the equilibrium
experiments.

252        The patterns of SSS difference between Exps MH and PI are similar to those of SST difference. In general, the

North Atlantic is more saline in Exp MH than in Exp PI (Fig. 8d), mainly due to stronger evaporation over
precipitation in Exp MH than in Exp PI (Fig. 12d), which is in turn due to the warmer SST forced by the ORB effect
(Fig. 8e). The polar oceans are fresher in Exp MH than in Exp PI (Figs. 8d, e), mainly due to more freshwater flux
coming from sea ice in Exp MH (Figs. 12a, b), consistent with the warmer climate in the MH due to the ORB effect.
The SSS difference caused by the GHG effect is roughly opposite to that caused by the ORB effect, but with much
weaker magnitude (Fig. 8f), because the cooling effect of the GHG makes less evaporation in the subtropical–subpolar
Atlantic and more sea ice in the polar oceans (Fig. 12c). Similar to the equilibrium experiments, the SSS changes in
the transient experiments show similar characteristics (Figs. 9d, e, f).

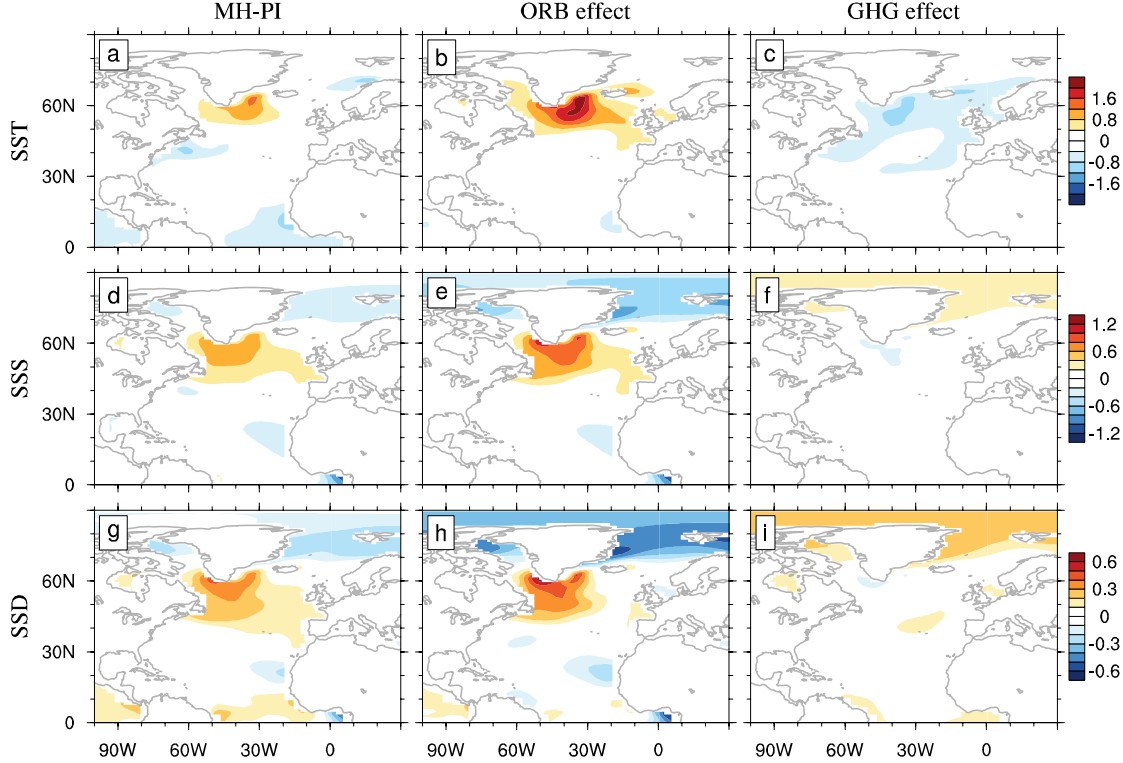

**Figure 8** Changes in (a)–(c) sea-surface temperature (SST), (d)–(f) sea-surface salinity (SSS), and (g)–(i) sea-surface density (SSD) of the North Atlantic in Exp MH, with respect to Exp PI. (a), (d), and (g) are for the total changes; (b), (e), and (h), for the changes due to ORB effect; (c), (f), and (i), for changes due to GHG effect. Units: °C for SST, psu for SSS, and kg/m³ for SSD.

The patterns of SSD difference (Figs. 8g–i) resemble those of both SSS and SST differences, while its polarity is determined by SSS difference. The higher SSD in the North Atlantic is favorable for a stronger deep-water formation and thus a stronger AMOC in Exp MH. Forced by the ORB effect, the North Atlantic surface ocean can be 0.5 kg/m³ denser in Exp MH than in Exp PI (Fig. 8h), which could have resulted in a 1.2-Sv stronger AMOC in Exp MH than in Exp PI (Fig. 6e). However, the GHG effect, although weak, has an opposite effect on SSD and thus the AMOC (Fig. 8i), and eventually mitigates the ocean change in Exp MH. Similar patterns of SSD are shown in the transient experiments, with increased North Atlantic density in Exp ORB, and the opposite and weaker effect in Exp GHG (Figs. 9g, h, i), corresponding to changes in the AMOC (Fig. 6). These suggest that the mechanisms of ORB and GHG on climate change in the equilibrium and transient experiments are consistent.

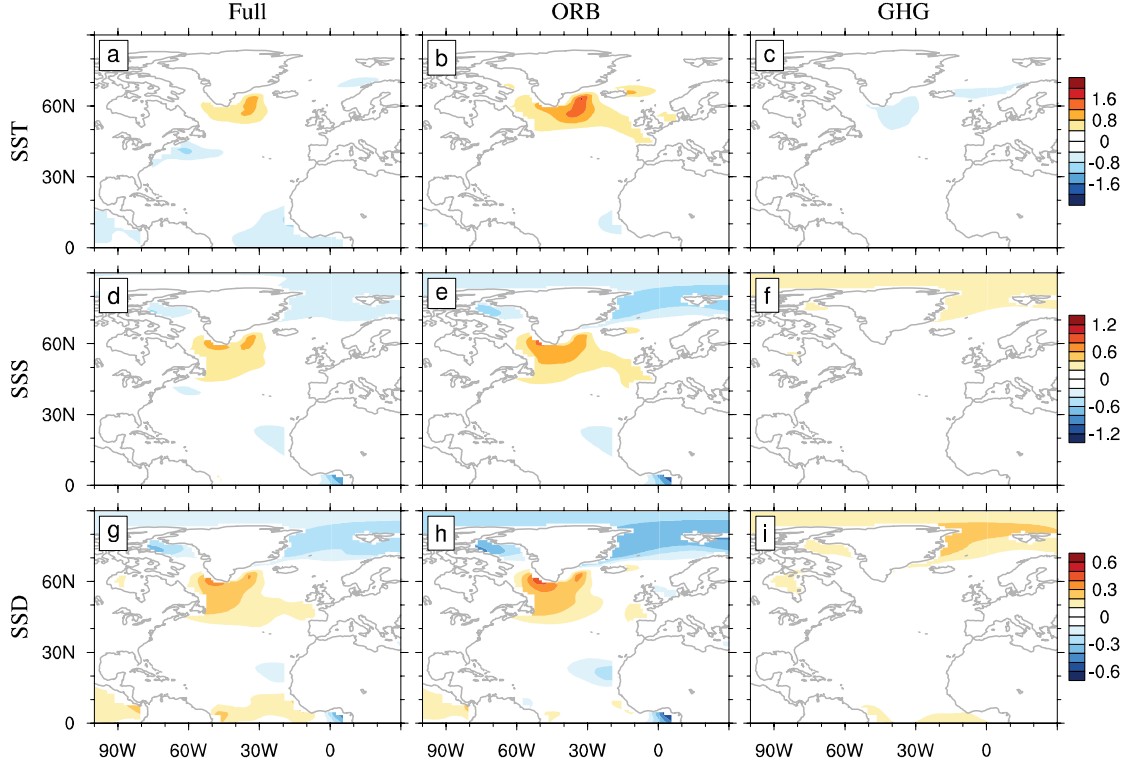

276

**Figure 9**  Similar to Fig. 8, but for Exps Full, ORB, and GHG, respectively. All variables represents changes between the two stages (Stage1-Stage2).

**4.2 Change in surface freshwater flux**

Sea-surface freshwater flux includes both sea-ice formation (melting) and net evaporation (i.e., evaporation minus precipitation, or EMP). Figure 10 shows the change of annual mean sea-ice thickness in the Arctic. The Arctic sea-ice thickness in Exp MH is about 1.0 m thinner than that in Exp PI (Fig. 10a). The largest sea-ice difference, which is about 3.0 m thinner in Exp MH, occurs in the Baffin Bay. When forced by the ORB effect only, the Arctic sea ice would be more than 1.5 m thinner (Fig. 10b), consistent with the stronger insolation and the warming in the NH high latitudes (Figs. 1, 2e). The GHG effect leads to a slight increase of sea ice in the Arctic (Fig. 10c) in Exp MH, which is less than 0.5 m in thickness. Changes in Arctic sea-ice thickness can affect sea ice transport to the subpolar Atlantic. The loss of sea ice in the central Arctic Ocean can reduce its export through the Fram Strait, which can lead to an increase in salinity in the associated subpolar [no "-"] regions (Shi and Lohmann, 2016), as shown in Figs. 8d and e. Similar changes in sea-ice thickness also occur in the transient experiments: the Arctic sea-ice thickness is decreased significantly in Exp ORB, while it is nearly unchanged in Exp GHG (Figs. 10e, f), reflecting the consistency of the effects of ORB and GHG in both equilibrium and transient experiments.

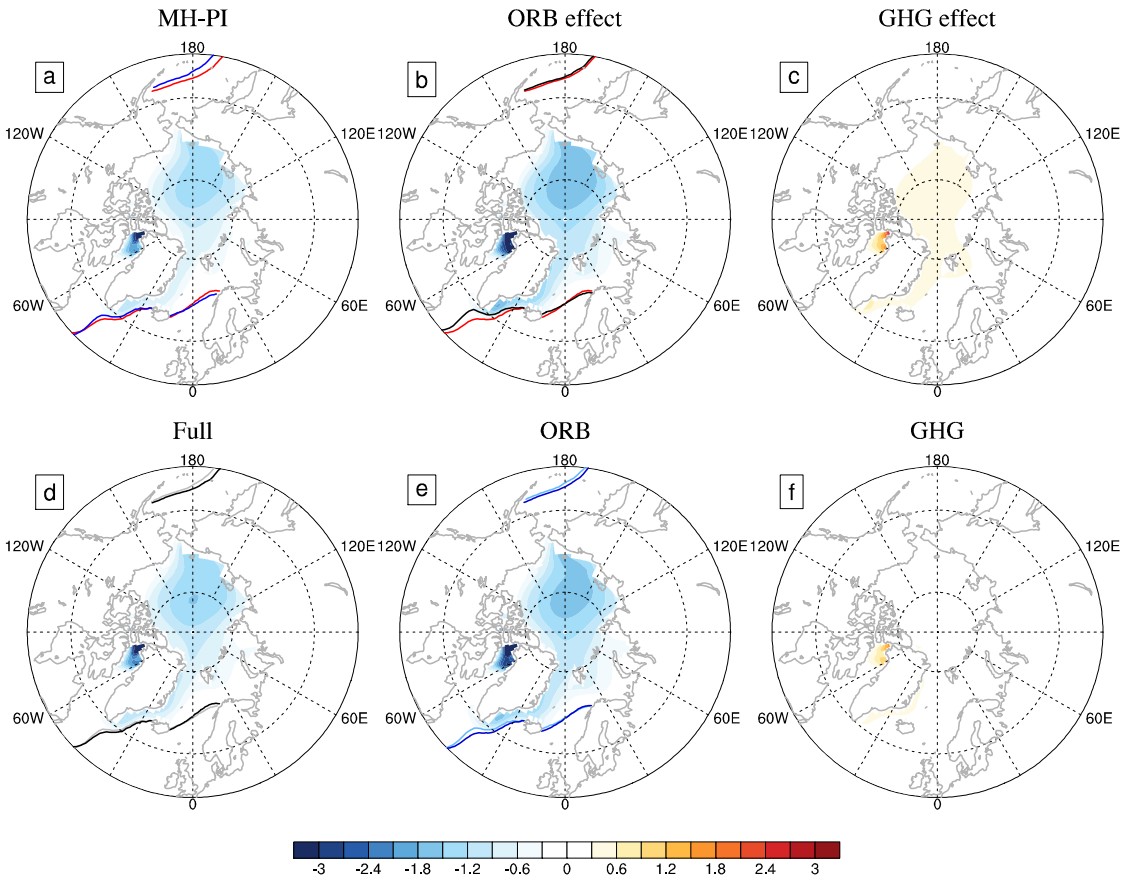

**Figure 10 (a)–(c) Changes in Arctic mean sea-ice thickness in Exp MH, with respect to Exp PI. Positive (negative) value represents sea-ice formation (melting). (a) is for the total change; (b) and (c), for changes due to ORB and GHG effects, respectively. Solid blue, black, and red curves show the sea-ice margin in Exps MH, MH_ORB and PI, respectively. (d)–(f) Same as (a)–(c), except for Exps Full, ORB, and GHG, respectively. The solid gray and light blue curves indicate the sea-ice margin of Stage1 in Exps Full and ORB, respectively; and the black and dark blue solid curves represent the sea-ice margin of Stage2 in Exps Full and ORB, respectively. The sea-ice margin is defined by the 15% sea-ice fraction. Units: m.**

The sea-ice margin in the North Atlantic in Exp MH is slightly more northward compared to that in Exp PI (solid blue curve, Fig. 11a). The curves in Fig. 11 show sea-ice margin in different experiments. The northward displacement of sea-ice margin and the decrease in sea-ice volume in the Arctic favor the decrease in freshwater flux in the North Atlantic, helping a more saline North Atlantic, which contributes about 0.9 psu 10yr$^{-1}$ to the SSS tendency between 40° and 60°N (Fig. 11a). The EMP flux is small, and the upper ocean is refreshed at a steady rate of about 0.09 psu 10 yr$^{-1}$ in the North Atlantic (Fig. 11d). The contributions of sea-ice change and EMP flux to SSS in the transient experiments are also about 0.9 and 0.09 psu 10 yr$^{-1}$, respectively (Figs. 11g, j). Overall, for the North Atlantic the change of sea ice plays a dominant role; and its contribution to SSS tendency is about 10 times that of EMP.

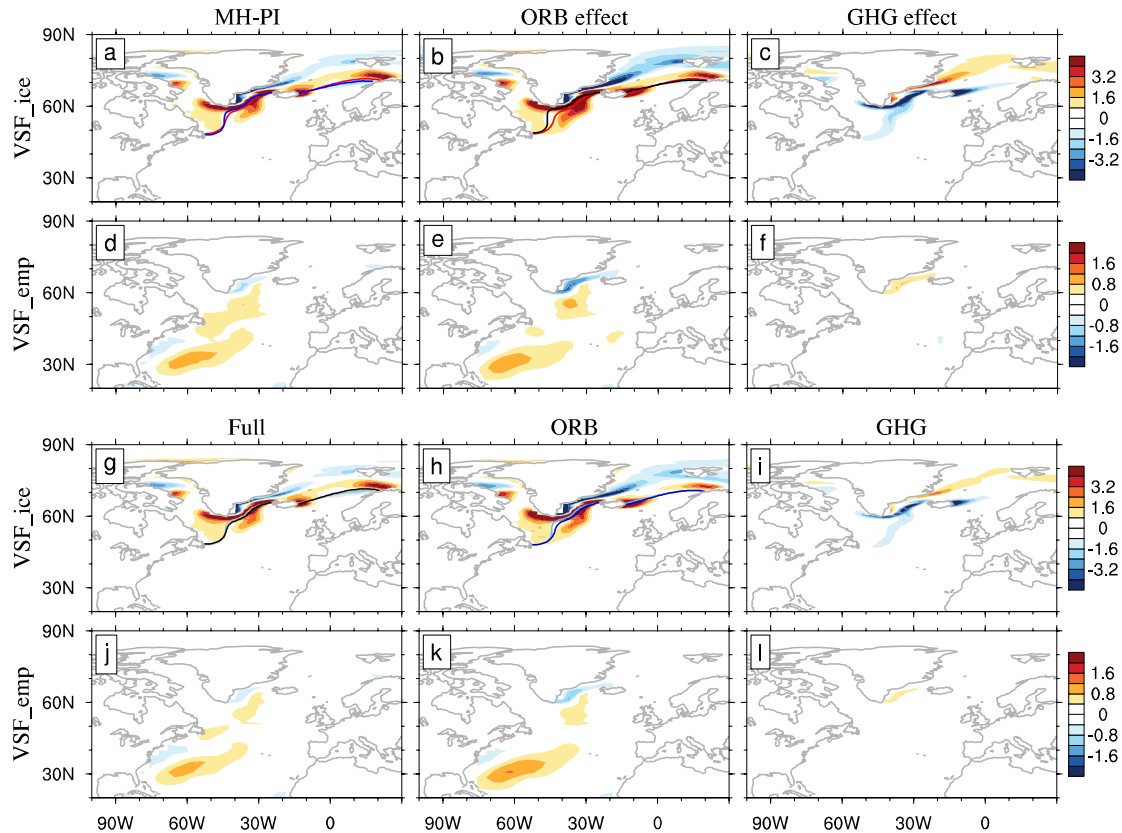

**Figure 11** **Changes in (a)–(c), namely, virtual salt flux (VSF) due to sea ice, and in (d)–(f), VSF due to EMP in Exp MH, with respect to Exp PI. Positive (negative) value represents sea-ice formation (melting) or evaporation larger (smaller) than precipitation. (a) and (d) are for total changes; (b) and (e), for changes due to ORB effect; (c) and (f), for GHG effect. (g)–(l) Same as (a)–(f), but for Exp Full, Exp ORB, and Exp GHG, respectively. The solid gray and light blue curves indicate the sea-ice margin of Stage1 in Exps Full and ORB, respectively; and the black and dark blue solid curves represent the sea-ice margin of Stage2 in Exps Full and ORB, respectively. The sea-ice margin in (a)–(b) is defined the same way as that in Fig. 8. Units: psu 10 yr$^{-1}$.**

The sea-ice margin in Exp MH is controlled by the ORB effect. In individual forcing experiment, the sea-ice margin forced by the ORB effect is almost the same as that in Exp MH (solid black curve, Fig. 11b). The contributions of ORB and GHG effects to changes in virtual salt flux (VSF) due to sea ice are 1.3 and -0.4 psu 10 yr$^{-1}$, respectively (Figs. 11b, c); and those due to the EMP flux are 0.06 and 0.03 psu 10 yr$^{-1}$, respectively (Figs. 11e, f). In the transient experiments, the contributions of ORB and GHG effects to the VSF due to sea ice are 1.1 and -0.2 psu 10 yr$^{-1}$, respectively (Figs. 11h, i); and those due to EMP flux are 0.05 and 0.03 psu 10 yr$^{-1}$, respectively (Figs. 11i, l). This suggests that the sea-ice change caused by the ORB effect plays an important role in the enhancement of the AMOC in Exp MH.

In general, the modelling results suggest that the stronger AMOC in the MH period resulted from more saline
North Atlantic, which was contributed mainly by smaller freshwater flux coming from the Arctic. The contribution of
EMP to salinity change was small, which was only one-tenth of the sea-ice contribution. ORB and GHG consistently
play opposite roles in the deep-water formation of the subpolar Atlantic. Their combined effect resulted in little
change in the AMOC in the MH period, which is less than 1 Sv enhancement in both equilibrium and transient
experiments.

## 5.   Summary and discussion

In this study, six experiments using the CESM1.0 were conducted to quantify the contributions of ORB and GHG
effects to the MH climate. Most attention was paid to the AMOC; and the mechanism to the insignificant difference of
the AMOC between the MH and PI periods was explored. This study is the first attempt to separate the ORB and GHG
effects on the MH climate. Simulations showed that the NH climate exhibits much greater regional and seasonal
variability due to the seasonal enhancement of insolation caused by changes in ORB; and these contrasting seasonal
responses lead to little change in annual mean climate. Lower GHG in Exp MH has a global cooling effect, with
greater temperature decreases at higher latitudes associated with feedbacks from sea ice and snow cover. The
combined effect of these two forcing factors leads to a weak warming at the NH high latitudes and cooling elsewhere,
similar to the temperature changes in the PMIP4 ensemble (Brierley et al., 2020).
Weakening meridional atmospheric temperature gradient in Exp MH leads to the Hadley cell being weakened by
about 10% in the NH. At the same time, due to the change of sea-surface buoyancy in the North Atlantic, the AMOC
is slightly enhanced by about 4%. As far as the changes in MHT magnitude in the NH are concerned, the effect of
ORB is about five times that of GHG. Our experiments also showed that the change in the AMOC is mostly
determined by the freshwater flux change in the North Atlantic, which is in turn closely related to the Arctic sea-ice
change related to the ORB effect. GHG has the opposite effect to ORB, which mitigates the enhancement of the
AMOC (Figs. 9b, c).
The conclusions drawn in this paper may be model-dependent. Shi and Lohmann (2016) simulated a stronger MH
AMOC in the high-resolution version of the ECHAM, with a maximum change of more than 2 Sv. Most of the models
in the CMIP5 reveal a positive AMOC change in the MH period. Some previous studies (Ganopolski et al., 1998;
Otto-Bliesner et al., 2006) showed that the AMOC in the MH is weaker than that of the PI period. The main reason for
the inconsistency is that the simulated ocean salinity in the North Atlantic is different. Therefore, it is necessary to
carry out simulations with multiple models to reduce model dependence. Our simulations of the AMOC in the MH are
similar to those of Jiang et al. (2023), both showing no significant changes in the AMOC in the MH compared with
the PI; however, their study did not explain the mechanism behind this phenomenon. Our study reveals the
competitive relationship between the two forcing factors through multiple-equilibrium state simulations and transient
simulations, supporting the popularconclusions about the AMOC change from the MH to the PI periods.

360        Our study focuses on the effects of ORB and GHG; and the simulated cooler annual mean temperature in most

areas of the NH differs from the warming record revealed by most proxy data (Wanner et al., 2008; Liu et al., 2014),
but is similar to the conclusions from the PMIP4 simulations. It is unclear whether these differences originate from the
model, the data record, or a combination of the two. Some proxy data suggested that the climate of North Africa was
wetter in the MH period, which was known as the Green Sahara (Larrosoana, 2012). Jiang et al. (2012) analyzed the
simulation results of six coupled models in PMIP2 for the MH period. They found that the dynamic vegetation effect
led to a decrease in annual cooling over China in five of these models during the MH period, although its impact on
the MH temperature was minimal. Braconnot et al. (2021) and Zhang et al. (2021a) studied the effect of dust reduction
on climate due to the greening of the Sahara desert, using the CESM and IPSL models, respectively, showing global
mean surface temperature increased by about 0.1°C. Although there are other forcing factors in the MH period, such
as vegetation, dust, and topography, overall our simulations are representative of the most important forcing factors
and provide quantified estimates of the contributions of ORB and GHG effects on the MH climate.

**Acknowledgements.** This work is supported by the National Natural Science Foundation of China (Nos. 42230403,
42288101, and 41725021) and by the foundation at the Shanghai Frontiers Science Centre of Atmosphere-Ocean
Interaction of Fudan University. The experiments were performed on the supercomputers at the Chinese National
Supercomputer Centre in Tianjin (Tian-He No.1).

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
