# Peer review of "Quantifying Effects of Earth Orbital Parameters and Greenhouse Gases on Mid-Holocene Climate"

_EGUsphere, 2023_

## Author Comment (AC3)

**Replies to Reviewer #1:**

*This paper describes the results of mid-Holocene simulations with CESM1 with a focus on separating the impacts of the altered orbit and lower CO2 relative to the pre-industrial. the authors also focus on the change in the AMOC. They find that the orbital forcing at 6kyr BP is much more important than the lower CO2 and that the AMOC strengthens as a result of sea-ice changes.*

*The text is well-written and clearly laid out. The main issue is in communicating where this paper fits into the established modelling literature and clearly articulating the advance here over other work.*

**Responses:** Thank you very much for your invaluable comments and suggestions. We will combine the comments of the two reviewers and focus on revising the following parts of the manuscript:

(1) In Section 1, aiming at the position of this paper in the established modeling literature, we will rewrite the introduction, comprehensively review the past research on coupled model simulation of AMOC in the mid-Holocene, discuss the insufficiency of previous studies, state the necessity of this research, and explain the mechanism of AMOC changes.

(2) In Section 2, we will provide a detailed description of our experimental setup, including the spin-up phase of the simulation and the resolution of our model. We will also explain the differences between our simulation and those carried out under the PMIP3 protocol, which is commonly used in paleoclimate modeling studies.

(3) In Section 3, we will increase the content of the time series of the simulated AMOC in order to better understand the changes in AMOC under different forcings. We will provide stronger evidence to support our conclusions by analyzing the time series data in detail.

(4) Some figures will be re-plotted to improve clarity.

**Main comments**

*Previous studies with multi-model ensembles have already shown that there is no model consensus on the AMOC at the mid-Holocene (Jiang et al., 2023) but also that the changes in AMOC are actually very small between the mid-Holocene and pre-industrial (e.g. figure 9 by Brierley et al 2020). Thus, this study's focus on AMOC could do with additional justification. This could be straightforward if the model used was significantly more advanced than those used before, but that is not the case. For example, Jiang et al. included CESM2 the successor of the model used here. Also, at T31gx3v7 (3 degrees in the ocean?) the present study is pretty coarse resolution. The changes in AMOC reported are also relatively small and might even fall close to the 0.95-1.05*

*uncertainty bound in Brierley et al (2020)'s figure 9. Given this context, the significance of the present findings needs to be clarified.*

**Responses:** Thank you very much for this comment. Recent studies on the AMOC during the mid-Holocene indicate that there was no significant change compared to the pre-industrial era. However, the mechanisms underlying this nearly unchanged AMOC remain unclear. Our paper makes two key contributions that distinguish it from other studies. Firstly, we separated the impact of two different forcings on the AMOC under the PMIP4 protocol. Secondly, we emphasize that changes in seasonality are much stronger and more important than changes in annual mean climate. Our study reveals the competitive relationship between the two forcings, which is a significant advancement. It supports the existing conclusion on the mid-Holocene AMOC and sheds light on the underlying mechanisms for the small differences observed during this period.

The decision to use a low-resolution version of CESM in this paper was based on two main reasons:

(1) Improved AMOC simulation: The low-resolution CESM demonstrates a more accurate simulation of the AMOC response in real-world conditions compared to the high-resolution version. Li et al. (2021) (AMOC Stability and Diverging Response to Arctic Sea Ice Decline in Two Climate Models. *J. Climate*, 34, 5443-5460) found that the low-resolution model exhibits better representation of the Atlantic salinity distribution, AMOC mean strength, and AMOC stability, suggesting it could be more realistic for studying the AMOC response to Arctic sea-ice decline. They stated that "A question arises as to which type of AMOC response would occur in nature: a strong AMOC weakening that leads to a new equilibrium state or a modest transient weakening followed by full recovery? Perhaps it is the former type of response since the low-resolution model is more realistic in that it has a better representation of the Atlantic salinity distribution, the AMOC mean strength and AMOC stability."

(2) Limited computing resources: Due to the long simulation times required for the experiments (2000 years each, totaling 6000 years), and the constraints of available computing resources, the low-resolution CESM was chosen as it is more resource-efficient while still addressing the research problem effectively.

**Technical corrections:**

*1. Line 69: Probably worth stating that this is ~3.75 degree resolution?*

**Responses:** Thank you very much for this suggestion. we will rewrite the sentence. "The atmospheric model has 26 vertical levels and T31 horizontal resolution (3.75°×3 3.75°). "

2. *Line 70: What does "gx3v7 horizontal resolution" mean?*

**Responses:** The model uses the T31_gx3v7 grid and the marine module POP2 uses the gx3v7 grid, which has 60 vertical levels, and a uniform 3.6°spacing in the zonal direction. In the meridional direction, the grid is nonuniformly spaced. It is 0.6°near the equator, gradually increasing to the maximum 3.4°at 35N/S and then decreasing poleward.

3. *Line 72: Not sure that Smith & Gregory, 2009 should be here?*

**Responses:** Thank you very much for this suggestion. We have removed this and replaced with Smith et al. (2010).

Smith, R. D., and Coauthors, 2010: The Parallel Ocean Program (POP) reference manual. Tech. Rep. LAUR-10-01853, Los Alamos National Laboratory, 140 pp.

4. *Lines 74-79: replace "in the X period" with "from the X period".*

**Responses:** Thank you very much for this suggestion. we have replaced all.

5. *Figure 5: Panel (f) is not described in the caption. Titles above panels d)-f) would improve clarity. Also, timeseries of the AMOC and some quantification of interannual to centennial variability in the model would help to assess the significance of the changes shown here.*

**Responses**: Thank you very much for these suggestions. We will correct errors and add titles above the panels. Timeseries of the AMOC in the three simulations will be added to the manuscript.

[Figure]

**Figure R1** Patterns of mean AMOC in (a) Exp MH, (b) Exp MH_ORB, and (c) Exp PI; and (d) the AMOC change in Exp MH, with respect to Exp PI. (e) and (f) show the AMOC changes due to the ORB effect and GHG effect, respectively. The AMOC index is defined as the maximum streamfunction in the range of 0–2000 m of 20°–70°N in the North Atlantic. Units: Sv.

6. *Figure 6: There are no dashed or dotted lines in the figure? Also I think the vertical scale in panels b and c should be the same.*

**Responses:** Thank you very much for these suggestions. We will re-plot Figure 6. Solid, dashed, and dotted for MH, MH_ORB, and PI, respectively, which are nearly overlapped.

[Figure]

7. *Line 291: This sounds a bit sceptical, you could say "most proxy data" and cite e.g. Larrosoana et al (2012) or similar.*

**Responses:** Sorry, we will rewrite this sentence.

8. *Line 291: the Jiang et al. study is about China so good to specify that here and please correct the brackets.*

**Responses:** Sorry, we will rewrite this sentence.

**Replies to Reviewer #2:**

*I find it hard to recommend that this article be published in Climate of the Past. This is not because the science, and especially the simulations, are wrong – I see little incorrect with them. But rather because I do not feel it represents a sufficient addition to the discipline to warrant publication. I consider this work to be more like the level of a (good) dissertation instead of being worthy a peer-reviewed article. The main reason for this arises from a lack of engagement with the existing literature and explanation of this study's contribution to it.*

*This work documents a series of simulation with the low-resolution version of CESM1. They consist of a preindustrial control run, a mid-Holocene simulation using the PMIP4 protocol and a mid-Holocene simulation using the PMIP3 protocol (although they are not called this). The large scale features of these simulations are presented in a logical fashion, and the orbital and greenhouse gas impacts are estimated. The main findings relate to compensating forced changes in AMOC resulting in little change under PMIP4 protocols. These are associated with changes in buoyancy flux.*

**Responses:** Thank you very much for your invaluable comments and suggestions. We will combine the comments of the two reviewers and focus on revising the following parts of the manuscript:

(1) In Section 1, aiming at the position of this paper in the established modeling literature, we will rewrite the introduction, comprehensively review the past research on coupled model simulation of AMOC in the mid-Holocene, discuss the insufficiency of previous studies, state the necessity of this research, and explain the mechanism of AMOC changes.

(2) In Section 2, we will provide a detailed description of our experimental setup, including the spin-up phase of the simulation and the resolution of our model. We will also explain the differences between our simulation and those carried out under the PMIP3 protocol, which is commonly used in paleoclimate modeling studies.

(3) In Section 3, we will increase the content of the time series of the simulated AMOC in order to better understand the changes in AMOC under different forcings. We will provide stronger evidence to support our conclusions by analyzing the time series data in detail.

(4) Some figures will be re-plotted to improve clarity.

1.  *Seen in the context of the existing literature, I have strong doubts that these findings are generalisable. Firstly, the AMOC has large internal variability – this even not acknowledged in the piece, let alone investigated.*

**Responses:** Thank you very much for your comments. This manuscript delves into the mechanisms responsible for the subtle differences in AMOC between two equilibrium experiments, while quantifying the effects of various forcings. Figure R2 displays a time series of AMOC across the three experiments. To complement our findings, we will include a discussion on the internal variability of AMOC in the manuscript.

2. *Secondly, it can take a long time for AMOC to equilibrate – my own simulations using this particular model shown millennial response times (Brierley & Fedorov, 2016).*

**Responses:** Thank you very much for your comments. Our three equilibrium experiments are each conducted over a 2000-year period. Figures R2 display the time series of global mean surface temperature and AMOC. The criteria for an equilibrium state are determined by the global mean surface temperature trend (< ±0.05 °C per century) and a stable AMOC (Zhang et al., 2021). It is evident that our simulation has reached an equilibrium state.

[Figure]

**Figure R2**      Evolutions of (left panels) annual mean globally mean surface temperature (units: K) and (right panels) the AMOC (units: Sv) in three experiments.

3. *Thirdly, if the findings are valid, then one would anticipate a robust AMOC in the PMIP3 ensemble that disappears in PMIP4. But this is not what is seen (Brierley et al, 2020).*

**Responses:** Thank you for the advices. First, we would like to emphasize that our experiment does not involve a comparison of AMOC during the mid-Holocene under the PMIP3 and PMIP4 protocols. Instead, we separated the impacts of orbital parameters and greenhouse gases within the PMIP4 framework. Our model sets the solar constant at 1360.75 W/m$^2$, while the PMIP3 protocol uses a solar constant of 1365 W/m$^2$. Additionally, there are minor differences in greenhouse gas concentrations between the two protocols (Kageyama et al., 2017).

Second, we totally agree with the notion that the AMOC differences between the two periods under the PMIP3 or PMIP4 frameworks are relatively small. Our research aims to further elucidate the mechanisms behind this phenomenon and quantify the influence of each forcing. This work serves as a supplement to previous research and provides additional support for earlier findings.

*I do not doubt that are sufficient data from the simulations to support research of a publishable level. But it will require (a) substantial new analysis, that is (b) focused on a novel research question and is (c) comprehensively placed in the context of the existing literature.*

**Second round reply to Reviewer #2:**

*I would like to thank the authors for outlining so clearly how they would improve this. Even with these improvements, I have strong doubts whether the revised manuscript would warrant publication in Climate of the Past. This is because I still find it hard to see what novel insight would be evidenced and conveyed by it.*

**Responses:** Thank you very much for the comments. We would like to emphasize that in our study, there are at least three novel insights:

(1) The ORB effect exhibits a stronger seasonal cycle in the mid-Holocene (MH) than in the pre-industrial era (PI), as illustrated in Fig. 1. Consequently, changes in seasonal climatology are more pronounced in the MH than in the PI. More importantly, these seasonal alterations are considerably more significant than changes in the annual mean climate. We believe that from the perspective of Earth's climate habitability, discussing shifts in seasonality is more meaningful than changes in the annual mean climate, a subject which has received insufficient attention in previous studies. Naturally, seasonal changes in atmospheric variables are significant, but for oceanic variables, particularly the AMOC, we can only discuss them in terms of the annual mean.

(2) The climate difference patterns between the MH and PI (as shown in Fig. 2) display a characteristic of polar amplification. Superficially, this resembles the polar amplification observed in the long-term trend of current climate change. However, the causes are different: the former results from orbital forcing, while the latter arises from global warming associated with greenhouse gases. Despite the different causes, the mechanisms of polar amplification might be similar, and thus the consequences of polar amplification in the MH could provide insights for predicting the effects of polar amplification in current climate change. These questions warrant further studies, and this manuscript provides a starting point.

(3) We have isolated the impact of two different forcings on the AMOC under the PMIP4 protocol, aiming to identify the mechanism driving the changes in the AMOC. The extent of AMOC change, whether strong or weak, is not the key question we are seeking to address.

*If this manuscript can convincingly demonstrate that the changes in orbital forcing experienced during in the mid-Holocene would drive an enhanced AMOC, then that would be worthy of publication as it overturns the prevailing opinion. However, I find it hard to see that being possible from the experiment that has performed. Firstly, this is because previous work has shown AMOC changes to be rather model dependent (e.g. Jiang et al, 2023, shows a decreasing*

*AMOC in EC-Earth, whilst Otto-Bliesner et al, 2020, shows it increasing in CESM2) – but only a single model is deployed here.*

**Response**: Thank you very much for your comments. We do not aim to challenge the prevailing consensus. Rather, we propose that the negligible difference in AMOC between the MH and PI could result from the counterbalancing effects of orbital forcing and greenhouse gas forcing, at least based on our model results. The orbital forcing alone leads to a marginally stronger AMOC in the MH compared to the PI, as shown in Fig. R2 of the previous response and now in Fig. R3.

We have also conducted a transient simulation (MH_ORB_transient), which starts from 6ka in the MH and integrates over 6000 years to the PI. In MH_ORB_transient, the $CO_2$ level is fixed at the MH level, and the orbital parameters shift from the MH to PI. The results show that under lower $CO_2$ levels, akin to those in the MH, the AMOC exhibits a clear weakening trend (as shown by the green curves in Fig. R3). This suggests that, under the influence of orbital forcing only, the AMOC in the PI should be slightly weaker, which is consistent with our equilibrium experiments MH_ORB and PI.

[Figure]

**Figure R3**    Evolutions of the AMOC (units: Sv) in experiments MH_ORB (black and red line), PI (grey and blue lines) and MH_ORB_transient (green lines).

Regarding the issue of model dependency, it's important to note that any scientific problem approached through modeling will inevitably face this concern. We can treat a model result as a kind of virtual reality and endeavor to understand the mechanisms that govern this simulated environment. The aim of this study is not to depict the absolute climate reality of the MH and PI eras, but rather to provide insight into the possible reasons why the differences between the two climate realities in the MH and PI periods are minimal.

*The new figure clearly demonstrates that the runs are sufficiently spun-up, but also show the relatively large amount of internal variability in AMOC within the simulations. Previous authors have already concluded that changes in AMOC seen in mid-Holocene simulations can arise from internal variability, rather be a forced response (Williams et al, 2020).*

**Response**: Thank you very much for your comments. The internal variability of the AMOC in our experiments is approximately within 10% of the mean state of AMOC (i.e., 20±2 Sv). We believe this level of internal variability to be quite reasonable.

We greatly appreciate your reference to the study by Williams et al. (2020). We agree that it's plausible that "changes in AMOC seen in mid-Holocene simulations can arise from internal variability, rather than being a forced response". However, this conclusion was also derived from one specific model (HadGEMs) and could thus be highly model-dependent. We value the contribution of Williams's study, as it provides a novel perspective that encourages consideration of the relationship between internal variability and changes in the mean state, an aspect that has largely been overlooked in previous studies.

*In my mind, internal variability is a simpler and more plausible explanation for the AMOC behaviors seen across the 3 simulations. This would not require invoking a previously undescribed response to orbital forcing compensated by a response to greenhouse gas forcing that operates in the opposite direction than that seen in future projections and assessed by the IPCC. Any revised manuscript would need to comprehensively disprove this simpler explanation, and I do not see how that could be done.*

**Response**: Thank you very much for your comments. You've raised an intriguing question: how to determine the significance of changes in the mean state, given the presence of strong internal variability within the system. A straightforward approach might involve considering the ratio of signal to noise (RSN). We can regard internal variability as the noise, and the change in the mean state as the signal. The RSN can be large when the external forcing is strong or the timeframe of the variable is long.

We fully agree that when the timeframe is short, the RSN will be very small, and thus, internal variability can significantly affect the behavior of the mean state. To enhance the RSN, one practical solution would be to extend the model experiments for additional thousands of years, until the internal variability can be roughly treated as white noise.

*Under an experimental setup where one forced response is computed as a residual, you would naturally expect to infer a compensation mechanism if internal variability resulted in the sensitivity experiment having a higher AMOC. Using a more sophisticated experimental design with more simulations (such as Lunt et al, 2021) could more robustly deconvolve the various forced responses.*

**Response**: Thank you very much for your comments. The approach of separating the effects of different forcings is a standard procedure when data is unavailable, inaccurate, or computational resources are limited. We are certainly open to considering the more sophisticated experimental design that you mentioned.

When researchers estimate the meridional heat transport (MHT) of the Earth's climate system, the combined MHT by both the atmosphere and ocean is typically calculated first, as it only requires the net shortwave flux and longwave flux at the top of the atmosphere. These values can be accurately obtained from observations. If sufficient atmospheric data is available, the atmospheric meridional heat transport (AHT) can also be directly calculated. The oceanic heat transport (OHT) can then be obtained as a residual. Alternatively, if atmospheric data is not accurate and net flux data on the ocean surface is available, the OHT can be calculated directly using the net surface flux, and the AHT can be obtained as a residual. We would like to emphasize that this practice of directly calculating one variable and obtaining another as a residual is quite common in many areas of climate research.

References:

Jiang, Z., Brierley, C., Thornalley, D., and Sax, S.: No changes in overall AMOC strength in interglacial PMIP4 time slices, Clim. Past, 19, 107–121, https://doi.org/10.5194/cp-19-107-2023, 2023.

Lunt, D. J., Chandan, D., Haywood, A. M., Lunt, G. M., Rougier, J. C., Salzmann, U., Schmidt, G. A., and Valdes, P. J.: Multi-variate factorisation of numerical simulations, Geosci. Model Dev., 14, 4307–4317, https://doi.org/10.5194/gmd-14-4307-2021, 2021.

Otto‑Bliesner, B.L., Brady, E.C., Tomas, R.A., Albani, S., Bartlein, P.J., Mahowald, N.M., Shafer, S.L., Kluzek, E., Lawrence, P.J., Leguy, G. and Rothstein, M., 2020. A comparison of the CMIP6 midHolocene and lig127k simulations in CESM2. Paleoceanography and Paleoclimatology, 35(11), p.e2020PA003957.

Williams, C. J. R., Guarino, M.-V., Capron, E., Malmierca-Vallet, I., Singarayer, J. S., Sime, L. C., Lunt, D. J., and Valdes, P. J.: CMIP6/PMIP4 simulations of the mid-Holocene and Last Interglacial using HadGEM3: comparison to the pre-industrial era, previous model versions and proxy data, Clim. Past, 16, 1429–1450, https://doi.org/10.5194/cp-16-1429-2020, 2020

---

## Author Response (AR1)

**Replies to the first-round's reviews:**

**Replies to Reviewer #1:**

*This paper describes the results of mid-Holocene simulations with CESM1 with a focus on separating the impacts of the altered orbit and lower CO2 relative to the pre-industrial. the authors also focus on the change in the AMOC. They find that the orbital forcing at 6kyr BP is much more important than the lower CO2 and that the AMOC strengthens as a result of sea-ice changes.*

*The text is well-written and clearly laid out. The main issue is in communicating where this paper fits into the established modelling literature and clearly articulating the advance here over other work.*

**Responses:** Thank you very much for your comments and suggestions. We have combined the comments of both reviewers and revised the manuscript as follows:

(1) In Section 1, considering the existing modeling literatures, we rewrote the introduction, comprehensively reviewed the past research on coupled model simulations of the AMOC in the mid-Holocene, discussed the weakness of these studies, stressed the necessity of this current research, and identified the mechanism of AMOC changes.

(2) In Section 2, we provide a detailed description of our experimental setup, including the spin-up phase of the simulation and the resolution of our model. We also explain the differences between our simulation and those carried out under the PMIP3 protocol, which are commonly used in paleoclimate modeling studies. We added three transient experiments, to compare with the equilibrium experiments. The time series of the AMOC simulated by different experiments is presented to better understand the variation of the AMOC under different forcings.

(3) In Sections 3 and 4, we compared the AMOC results from both equilibrium and transient simulations, to strengthen our conclusions.

(4) Some figures were re-plotted to improve their clarity.

**Main comments**

*Previous studies with multi-model ensembles have already shown that there is no model consensus on the AMOC at the mid-Holocene (Jiang et al., 2023) but also that the changes in AMOC are actually very small between the mid-Holocene and pre-industrial (e.g. figure 9 by Brierley et al 2020). Thus, this study's focus on AMOC could do with additional justification. This could be straightforward if the model used was significantly more advanced than those used before,*

*but that is not the case. For example, Jiang et al. included CESM2 the successor of the model used here. Also, at T31gx3v7 (3 degrees in the ocean?) the present study is pretty coarse resolution. The changes in AMOC reported are also relatively small and might even fall close to the 0.95-1.05 uncertainty bound in Brierley et al (2020)'s figure 9. Given this context, the significance of the present findings needs to be clarified.*

**Responses:** Thank you very much for this comment. Recent studies on the AMOC during the mid-Holocene indicate no significant change compared to the pre-industrial era. However, the mechanism underlying this nearly unchanged AMOC remains unclear. Our study makes two key contributions that distinguish it from the other studies. First, we separate the impact of two different forcing factors on the AMOC under the PMIP4 protocol. Second, we emphasize that changes in seasonality are much stronger and more important than changes in annual mean climate. We reveal the competitive relationship between the two forcing factors, which is a significant advancement: It supports the existing conclusion on the mid-Holocene AMOC and sheds light on the underlying mechanisms for the small differences observed during this period.

The decision to use a low-resolution version of the CESM in this paper was based on two main considerations:

(1) Improved AMOC simulation: The low-resolution CESM gives a more accurate simulation of AMOC response in real-world conditions compared to the high-resolution version. Li et al. (2021) (AMOC Stability and Diverging Response to Arctic Sea Ice Decline in Two Climate Models. *J. Climate*, 34, 5443-5460) found that the low-resolution model has a better representation of Atlantic salinity distribution, AMOC mean strength, and AMOC stability, suggesting such model be more realistic for studying AMOC response to Arctic sea-ice decline. They stated that "A question arises as to which type of AMOC response would occur in nature: a strong AMOC weakening that leads to a new equilibrium state or a modest transient weakening followed by full recovery? Perhaps it is the former type of response since the low-resolution model is more realistic in that it has a better representation of the Atlantic salinity distribution, the AMOC mean strength and AMOC stability."

(2) Limited computing resources: Due to the long simulation times required for the experiments (2000 years each, totaling 6000 years) and the constraints of available computing resources, we chose the low-resolution CESM as it is more resource-efficient while still addressing the research problem effectively.

**Technical corrections:**

*1. Line 69: Probably worth stating that this is ~3.75 degree resolution?*

**Responses:** Thank you very much for this suggestion. we rewrote the sentence. "The atmospheric model has 26 vertical levels and T31 horizontal resolution (3.75°×3 3.75°). "

*2. Line 70: What does "gx3v7 horizontal resolution" mean?*

**Responses:** The model uses the T31_gx3v7 grid; and the ocean model component POP2 uses the gx3v7 grid, which has 60 vertical levels, and a uniform 3.6° spacing in the zonal direction. In the meridional direction, the grid is nonuniformly spaced: It is 0.6° near the equator, gradually increasing to the maximum 3.4° at 35°N/°S and then decreasing poleward.

*3. Line 72: Not sure that Smith & Gregory, 2009 should be here?*

**Responses:** Thank you very much for this suggestion. We removed this and replaced with Smith et al. (2010).

Smith, R. D., and Coauthors, 2010: The Parallel Ocean Program (POP) reference manual. Tech. Rep. LAUR-10-01853, Los Alamos National Laboratory, 140 pp.

*4. Lines 74-79: replace "in the X period" with "from the X period".*

**Response:** Thank you very much for this suggestion. we replaced all.

*5. Figure 5: Panel (f) is not described in the caption. Titles above panels d)-f) would improve clarity. Also, timeseries of the AMOC and some quantification of interannual to centennial variability in the model would help to assess the significance of the changes shown here.*

**Responses**: Thank you very much for these suggestions. Figure 5 in the previous manuscript becomes Fig. 6 in the revised manuscript. The AMOC changes in the three transient simulations are also plotted in Fig. 6. Please refer to Fig. R1. We corrected errors and labeled each panel. Timeseries of the AMOC in all simulations are shown in Fig. 5 in the revised manuscript.

[Figure]

**Figure R1**    Patterns of mean AMOC in (a) Exp MH, (b) Exp MH_ORB, and (c) Exp PI; and (d) AMOC change in Exp MH, with respect to Exp PI. (e) and (f) show AMOC changes due to the ORB effect and GHG effect, respectively. (g, h, i) represent the changes of the two stages AMOC (stage1-stage2) in Exps Full, ORB, and GHG, respectively. The AMOC index is defined as the maximum streamfunction in the range of 0–2000 m of 20°–70°N in the North Atlantic. Units: Sv.

*6.    Figure 6: There are no dashed or dotted lines in the figure? Also I think the vertical scale in panels b and c should be the same.*

**Responses:** Thank you very much for these suggestions. Original Fig. 6 becomes Fig. 7 in the revised manuscript. Please refer to Fig. R2. Solid, dashed, and dotted for MH, MH_ORB, and PI, respectively, which are hard to see as they are nearly overlapped.

[Figure]

**Figure R2**      (a) Annual mean meridional heat transport (MHT). Black, red, and blue for the total MHT, AHT, and OHT, respectively. Solid, dashed, and dotted for Exps MH, MH_ORB, and PI, respectively. (b) and (c) show changes in the total MHT, AHT, and OHT due to ORB and GHG effects, respectively. Units: PW (1 PW = $10^{15}$ W).

7. *Line 291: This sounds a bit sceptical, you could say "most proxy data" and cite e.g. Larrosoana et al (2012) or similar.*

**Response:** Sorry. We fixed the error.

8. *Line 291: the Jiang et al. study is about China so good to specify that here and please correct the brackets.*

**Responses:** Sorry, we rewrote this sentence. "Jiang et al. (2012) analyzed the simulation results of six coupled models in PMIP2 for the mid-Holocene period. They found that the dynamic vegetation effect led to a decrease in annual cooling over China in five of these models during this period, although its impact on the mid-Holocene temperature was minimal."

**Replies to Reviewer #2:**

*I find it hard to recommend that this article be published in Climate of the Past. This is not because the science, and especially the simulations, are wrong – I see little incorrect with them. But rather because I do not feel it represents a sufficient addition to the discipline to warrant publication. I consider this work to be more like the level of a (good) dissertation instead of being worthy a peer-reviewed article. The main reason for this arises from a lack of engagement with the existing literature and explanation of this study's contribution to it.*

*This work documents a series of simulation with the low-resolution version of CESM1. They consist of a preindustrial control run, a mid-Holocene simulation using the PMIP4 protocol and a mid-Holocene simulation using the PMIP3 protocol (although they are not called this). The large scale features of these simulations are presented in a logical fashion, and the orbital and greenhouse gas impacts are estimated. The main findings relate to compensating forced changes in AMOC resulting in little change under PMIP4 protocols. These are associated with changes in buoyancy flux.*

**Responses:** Thank you very much for your critical comments and suggestions. We have combined comments of both reviewers and revised the manuscript as follows:

(1) In Section 1, considering the existing modeling literatures, we rewrote the introduction, comprehensively reviewed the past research on coupled model simulations of the AMOC in the mid-Holocene, discussed the weakness of these studies, stressed the necessity of this current research, and identified the mechanism of AMOC changes.

(2) In Section 2, we provide a detailed description of our experimental setup, including the spin-up phase of the simulation and the resolution of our model. We also explain the differences between our simulation and those carried out under the PMIP3 protocol, which are commonly used in paleoclimate modeling studies. We added three transient experiments, to compare with the equilibrium experiments. The time series of the AMOC simulated by different experiments is presented to better understand the variation of the AMOC under different forcings.

(3) In Sections 3 and 4, we compared the AMOC results from both equilibrium and transient simulations, to strengthen our conclusions.

(4) Some figures were re-plotted to improve their clarity.

1. *Seen in the context of the existing literature, I have strong doubts that these findings are generalisable. Firstly, the AMOC has large internal variability – this even not acknowledged in the piece, let alone investigated.*

**Responses:** Thank you very much for your comments. This manuscript delves into the mechanisms responsible for the subtle differences in the AMOC between two equilibrium experiments, while quantifying the effects of various forcing factors. Figure R3 displays the time series of the AMOC in the three experiments. To complement our findings, we included a discussion on the internal variability of the AMOC in the revised manuscript.

2. *Secondly, it can take a long time for AMOC to equilibrate – my own simulations using this particular model shown millennial response times (Brierley & Fedorov, 2016).*

**Responses:** Thank you very much for your comments. Each of our three equilibrium experiments was integrated over a 2000-year period. Figures R3 shows the time series of global mean surface temperature (GMST) and AMOC. The criteria for an equilibrium state are determined by the GMST trend (< ±0.05 °C per century) and a stable AMOC (Zhang et al., 2021). It is evident that each of our simulation has reached an equilibrium state.

[Figure]

**Figure R3**    Evolutions of (left panels) annual mean global mean surface temperature (GMST) (units: K) and (right panels) AMOC (units: Sv) in three experiments.

3. *Thirdly, if the findings are valid, then one would anticipate a robust AMOC in the PMIP3 ensemble that disappears in PMIP4. But this is not what is seen (Brierley et al, 2020).*

**Responses:** Thank you for the advice. First, we would like to emphasize that this study does not involve any comparison of the AMOC during the mid-Holocene under the PMIP3 and PMIP4 protocols. Instead, we separated the impacts of orbital parameters and greenhouse gases within the PMIP4 framework. We set the solar constant at 1360.75 W/m$^2$, while the PMIP3 protocol uses a solar constant of 1365 W/m$^2$. Additionally, there are minor differences in greenhouse gas concentrations between the two protocols (Kageyama et al., 2017).

Second, we totally agree that the AMOC differences between the two periods under the PMIP3 or PMIP4 frameworks are relatively small. Our research aims to further elucidate the mechanisms behind this phenomenon and quantify the influence of each forcing. This work serves as a supplement to previous research, and provides additional support for earlier findings.

*I do not doubt that are sufficient data from the simulations to support research of a publishable level. But it will require (a) substantial new analysis, that is (b) focused on a novel research question and is (c) comprehensively placed in the context of the existing literature.*

**Replies to the second-round's review:**

**Replies to Reviewer #2:**

*I would like to thank the authors for outlining so clearly how they would improve this. Even with these improvements, I have strong doubts whether the revised manuscript would warrant publication in Climate of the Past. This is because I still find it hard to see what novel insight would be evidenced and conveyed by it.*

**Responses:** Thank you very much for these comments. We would like to emphasize at least three novel insights in our study:

(1) The ORB effect exhibits a stronger seasonal cycle in the mid-Holocene (MH) than in the pre-industrial era (PI), as illustrated in Fig. 1. Consequently, changes in seasonal climatology are more pronounced in the MH than in the PI. More importantly, these seasonal alterations are considerably more significant than changes in the annual mean climate. We believe that from the perspective of Earth's climate habitability, discussing shifts in seasonality is more meaningful than discussing changes in the annual mean climate, a subject which has received insufficient attention in previous studies. Naturally, seasonal changes in atmospheric variables are significant, but for oceanic variables, particularly the AMOC, we can only discuss them in terms of the annual mean.

(2) The climate difference patterns between the MH and PI (as shown in Fig. 3 in the revised manuscript) display a characteristic of polar amplification. Superficially, this resembles the polar amplification observed in the long-term trend of current climate change. However, the causes are different: the former results from orbital forcing, while the latter arises from global warming associated with greenhouse gases. Despite the different causes, the mechanisms of polar amplification might be similar, and thus the consequences of polar amplification in the MH could provide insights for predicting the effects of polar amplification in current climate change. These questions warrant further studies; and this manuscript provides a starting point.

(3) We isolated the impact of two different forcing factors on the AMOC under the PMIP4 protocol, aiming to identify the mechanism driving the changes in the AMOC. The extent of AMOC change, whether strong or weak, is not the key question we intend to address here.

*If this manuscript can convincingly demonstrate that the changes in orbital forcing experienced during in the mid-Holocene would drive an enhanced AMOC, then that would be worthy of publication as it overturns the prevailing opinion. However, I find it hard to see that being possible from the experiment that has performed. Firstly, this is because previous work has*

*shown AMOC changes to be rather model dependent (e.g. Jiang et al, 2023, shows a decreasing AMOC in EC-Earth, whilst Otto-Bliesner et al, 2020, shows it increasing in CESM2) – but only a single model is deployed here.*

**Responses**: Thank you very much for your comments. We do not aim to challenge the prevailing consensus. Rather, we propose that the negligible difference in the AMOC between the MH and PI could result from the counterbalancing effects of orbital forcing and greenhouse gas forcing, at least based on our model results. The orbital forcing alone leads to a marginally stronger AMOC in the MH compared to the PI, as shown in Fig. R3 of the previous responses, and   in Fig. R4.

We also conducted a transient simulation (MH_ORB_transient), which started from 6ka in the MH and was integrated over 6000 years to the PI. In MH_ORB_transient, the $CO_2$ level is fixed at the MH level, and the orbital parameters shift from the MH to the PI. The results show that under lower $CO_2$ levels, akin to those in the MH, the AMOC exhibits a clear weakening trend (as shown by the green curves in Fig. R4). This suggests that, under the influence of orbital forcing alone, the AMOC in the PI should be slightly weaker, which is consistent with our equilibrium experiments MH_ORB and PI.

[Figure]

**Figure R4**    Evolutions of AMOC (units: Sv) in experiments MH_ORB (black and red lines), PI (grey and blue lines) and MH_ORB_transient (green line).

Regarding the issue of model dependency, it's important to note that any scientific problem approached through modeling will inevitably face this concern. We can treat model results as a kind of virtual reality and endeavor to understand the mechanisms that govern this simulated environment. The aim of this study is not to depict the absolute climate states of the MH and PI eras, but rather to provide insight into the possible reasons why the differences between the two climate states in the MH and PI periods are minimal.

*The new figure clearly demonstrates that the runs are sufficiently spun-up, but also show the relatively large amount of internal variability in AMOC within the simulations. Previous authors have already concluded that changes in AMOC seen in mid-Holocene simulations can arise from internal variability, rather be a forced response (Williams et al, 2020).*

**Responses**: Thank you very much for your comments. The internal variability of the AMOC in our experiments is approximately within 10% of the mean state of the AMOC (i.e., 20±2 Sv). We believe this level of internal variability be quite reasonable.

We greatly appreciate your reference to the study by Williams et al. (2020). We agree that it's plausible that "changes in AMOC seen in mid-Holocene simulations can arise from internal variability, rather than being a forced response". However, this conclusion was derived from one specific model (HadGEMs), and could thus be highly model-dependent. We value the contribution by Williams et al. (2020), as it provides a novel perspective that encourages consideration of the relationship between internal variability and changes in the mean state, an aspect that has largely been overlooked in previous studies.

*In my mind, internal variability is a simpler and more plausible explanation for the AMOC behaviors seen across the 3 simulations. This would not require invoking a previously undescribed response to orbital forcing compensated by a response to greenhouse gas forcing that operates in the opposite direction than that seen in future projections and assessed by the IPCC. Any revised manuscript would need to comprehensively disprove this simpler explanation, and I do not see how that could be done.*

**Responses**: Thank you very much for your comments. You've raised an intriguing question: how to determine the significance of changes in the mean state, given the presence of strong internal variability within the system. A straightforward approach may involve considering the ratio of signal to noise (SNR). We can regard internal variability as the noise, and the change in the mean state as the signal. The SNR can be large when the external forcing is strong or the timeframe of the variable is long.

We fully agree that when the timeframe is short, the SNR will be very small; and thus, internal variability can significantly affect the behavior of the mean state. To improve the SNR, one practical solution is to extend the model experiments for another thousands of years, until the internal variability can be roughly treated as white noise.

*Under an experimental setup where one forced response is computed as a residual, you would naturally expect to infer a compensation mechanism if internal variability resulted in the sensitivity experiment having a higher AMOC. Using a more sophisticated experimental design with more simulations (such as Lunt et al, 2021) could more robustly deconvolve the various forced responses.*

**Responses**: Thank you very much for your comments. The approach of separating the effects of different forcing factors is a standard procedure when data is unavailable, inaccurate, or computational resources are limited. We are certainly open to considering these more sophisticated experimental design that you mentioned.

When researchers estimate the meridional heat transport (MHT) of the Earth's climate system, the combined MHT by both the atmosphere and ocean is typically calculated first, as it only requires the net shortwave flux and longwave flux at the top of the atmosphere. These values can be accurately obtained from observations. If sufficient atmospheric data is available, the atmospheric meridional heat transport (AHT) can also be directly calculated. The oceanic heat transport (OHT) can then be obtained as the residual. Alternatively, if atmospheric data is not accurate and net flux data on the ocean surface is available, the OHT can be calculated directly using the net surface flux; and the AHT can be obtained as the residual. Note that this practice of directly calculating one variable and obtaining the other as the residual is quite common in climate research.

**References:**

Jiang, Z., Brierley, C., Thornalley, D., and Sax, S.: No changes in overall AMOC strength in interglacial PMIP4 time slices, Clim. Past, 19, 107–121, https://doi.org/10.5194/cp-19-107-2023, 2023.

Lunt, D. J., Chandan, D., Haywood, A. M., Lunt, G. M., Rougier, J. C., Salzmann, U., Schmidt, G. A., and Valdes, P. J.: Multi-variate factorisation of numerical simulations, Geosci. Model Dev., 14, 4307–4317, https://doi.org/10.5194/gmd-14-4307-2021, 2021.

Otto‑Bliesner, B.L., Brady, E.C., Tomas, R.A., Albani, S., Bartlein, P.J., Mahowald, N.M., Shafer, S.L., Kluzek, E., Lawrence, P.J., Leguy, G. and Rothstein, M., 2020. A comparison of the CMIP6 midHolocene and lig127k simulations in CESM2. Paleoceanography and Paleoclimatology, 35(11), p.e2020PA003957.

Williams, C. J. R., Guarino, M.-V., Capron, E., Malmierca-Vallet, I., Singarayer, J. S., Sime, L. C., Lunt, D. J., and Valdes, P. J.: CMIP6/PMIP4 simulations of the mid-Holocene and Last Interglacial using HadGEM3: comparison to the pre-industrial era, previous model versions and proxy data, Clim. Past, 16, 1429–1450, https://doi.org/10.5194/cp-16-1429-2020, 2020

**Replies to the third-round's review:**

**Replies to Reviewer #2:**

*If you were to include a new transient simulation from the mid-Holocene to present in your manuscript, then I would harbour no doubts about whether there is sufficient novelty in the manuscript to warrant publication.*

*In fact, we had a paper published in GRL that compiles the AMOC trends from a collection of transient simulations yesterday: https://agupubs.onlinelibrary.wiley.com/doi/10.1029/2023GL103078. If I'd have known about your run, we'd have invited you onboard as a co-author.*

**Responses**: Thank you very much for your comments and suggestions. We have included the transient runs in the revised manuscript, along with more analyses. Your new study in GRL is very helpful.

---

## Author Response (AR2)

**Replies to the fourth-round's reviews:**

**Replies to Reviewer #1:**

*Overall, I consider this revised version to be much improved. The addition of the new transient simulations definitely increases the novelty and robustness of the work. I would not be disappointed to see the manuscript published as is, but there are few things that I feel could be improved a little if there was an opportunity.*

*The mid-Holocene experiment in previous version of PMIP (particularly PMIP3) involved solely altering the orbital configuration. The manuscript currently does not describe that adequately, that all analysis of these simulations has effectively isolated the ORB component. The solar constant in PMIP3 was specified to be same value used for each model's preindustrial simulation, although that itself was not a single number across all the models. This is highlighted in Otto-Bliesner et al (2017), as it means that the local insolation changes can vary slightly in magnitude between the various PMIP3 experiments.*

**Responses:** Thank you very much for your comments and suggestions, we have added a sentence around line 91: "By tightly controlling the external forcings in the different experiments, our simulations effectively isolate the external forcing component compared to PMIP3, not just the ORB."

*I still feel that the issue of internal variability could be more explicitly addressed in the manuscript. However, the inclusion of the new simulations make be confident now that internal variability is not the explanation for the changes seen in the MH experiments. This could be achieved by, for example, adding a sentence or two, including error bars on Fig. 7, or stippling on Fig. 6 with a significance test (based on low frequency variability).*

**Responses:** Thank you very much for this comment, we have redrawn figure 6 with a significance test and added the sentence around line 205: "Some scholars have suggested that the change of AMOC in Exp MH may come from internal variability (Williams et al., 2020), but it is clear from our simulations that changes in response to external forcings are the main reason for the variations that occur in Exp MH (Fig. 6)."

[Figure]

**Figure 6** Patterns of mean AMOC in (a) Exp MH, (b) Exp MH_ORB, and (c) Exp PI; and (d) AMOC change in Exp MH, with respect to Exp PI. (e) and (f) show AMOC changes due to the ORB effect and GHG effect, respectively. g, h, i represent the AMOC changes between the two stages (Stage1-Stage2) in Exps Full, ORB, and GHG, respectively. The AMOC index is defined as the maximum streamfunction in the range of 0–2000 m of 20°–70°N in the North Atlantic. Stippling shows significance over the 90% level calculated by Student t-test. Units: Sv.

**Technical corrections:**

1. *L15: 'remarkedly' should be 'markedly'*     Revised.

2. *L79: your space is in the wrong place (before T31 rather than 26)*     Revised.

3. *L105: The sentence starting "Each transient…" would be better placed earlier in the paragraph.*     Revised.

4. *Around L240, you could mention comparison with the patterns seen in this experiment, and the 'warming hole' seen in observations.*

**Response:** Thank you very much for this suggestion, we have added the sentence around line 244: "This is in contrast to the warming hole shown by observations, which are dominated by the cooling of the North Atlantic in the context of global warming."

5. *L278. I do not understand what is meant by [no "-"]. I can see no dashes on Fig 10.*

**Responses**: Thank you very much for this suggestion, we have fixed the error.

6. *L314. A space is missing before 'In'*     Revised.

7. *L353. Add space into "studyfocuses"*     Revised.

8. *L357. I believe that the Larrosoana references would be better evidencing the statement that N Africa was wetter in the MH period.*     Revised.